# Chromosome fusions shaped karyotype evolution and evolutionary relationships in the model family Brassicaceae

Xinyao Jiang [1], Quanjun Hu[1], Dong Mei[1], Xiaonan Li [1], Ling Xiang [1], Ihsan A. Al-Shehbaz[2], Xiaoming Song[3], Jianquan Liu [1,4] ✉, Martin A. Lysak [5] ✉ & Pengchuan Sun [1,6] ✉

The ancestral crucifer karyotype and 22 conserved genomic blocks (CGBs) facilitate phylogenomic analyses in the Brassicaceae. Chromosomal rearrangements reshuffled CGBs of ancestral chromosomes during karyotype evolution. Here, we identify eight protochromosomes representing the common ancestral karyotype (ACBK) of the two Brassicoideae supertribes: Camelinodae (Lineage I) and Brassicodae (Lineage II). The characterization of multiple cascading fusion events allows us to infer evolutionary relationships based on these events. In the Camelinodae, the ACBK first evolved into the AKI genome, which remained conserved in the Cardamineae, whereas it was altered to tAKI by a reciprocal translocation that preceded the diversification of most Camelinodae tribes. The identified fusion breakpoints largely overlap with CGB boundaries, suggesting that CGBs are mainly disrupted by chromosome fusions. Our results demonstrate the stable inheritance of chromosome fusions and their importance for reconstructing evolutionary relationships. The chromosomal breakpoint approach provides a basis for ancestral state reconstruction based on chromosome-level genome assemblies.

The Brassicaceae family, comprising about 4000 species, has a global distribution and remarkable morphological diversity[1–3]. With their rich genomic resources at the chromosome level, the Brassicaceae stand out as an excellent model family for investigating the evolution of karyotypes since the origin of the family almost 40 million years ago[3]. The concept of the Ancestral Crucifer Karyotype (ACK), consisting of 8 chromosomes, was introduced by Schranz et al.[4]. The boundaries of 24 conserved genomic blocks (CGBs) were identified based on cross-species in situ hybridization of chromosome-specific BAC contigs of the model species *Arabidopsis thaliana* and comparative genetic mapping between *A. thaliana* and other crucifer species[4]; a later

revision led to the reduction of CGBs to 22[5]. Based on shared homologous CGBs, the ancestral karyotype of Lineage I (LI, Camelinodae *sensu*[6]) was proposed to be identical to the ACK, whereas the ancestral karyotype of Lineage II (LII, Brassicodae *sensu*[6]), known as the Proto-Calepineae Karyotype, was proposed to be derived from the ACK by descending dysploidy from $p = 8$ to $p = 7$[5,7] ($p$: inferred ancestral chromosome number[8]). The CGBs and inferred ancestral genomes have been widely used to reconstruct chromosome and genome evolution in the Brassicaceae[5,7,9–22].

Paleogenomics involves the inference of hypothetical ancestral karyotypes and the reconstruction of their evolution through non-

[1]Key Laboratory for Bio-resources and Eco-environment & Sichuan Zoige Alpine Wetland Ecosystem National Observation and Research Station, College of Life Sciences, Sichuan University, Chengdu, China. [2]Missouri Botanical Garden, St. Louis, MO, USA. [3]School of Life Sciences, North China University of Science and Technology, Tangshan, Hebei, China. [4]State Key Laboratory of Herbage Improvement and Grassland Agro-ecosystem, College of Ecology, Lanzhou University, Lanzhou, China. [5]CEITEC - Central European Institute of Technology and Department of Experimental Botany, Faculty of Science, Masaryk University, Brno, Czech Republic. [6]National Key Laboratory of Tropical Crop Breeding, Tropical Crops Genetic Resources Institute, Chinese Academy of Tropical Agricultural Sciences, Haikou, China. ✉e-mail: liujq@nwipb.ac.cn; martin.lysak@ceitec.muni.cz; sunpengchuan@gmail.com

dysploid (such as inversions and reciprocal translocations) or dysploid rearrangements of ancestral chromosomes. The latter lead either to an increase in the number of chromosomes (chromosome fission) or to a decrease (chromosome fusion)[23,24]. When exact chromosomal breakpoints are not known, the reduced number of linkage groups is generally interpreted as chromosome fusions. However, some of these events, despite appearing to have the same origin, may actually result from different mechanisms of descending dysploidy and/or have different breakpoints.

By identifying shared chromosome-like synteny blocks (CLSBs) as protochromosomes and tracing their dynamic evolution through three types of chromosomal rearrangements, we can uncover the sequence of chromosomal changes that have occurred from ancestral karyotypes to modern genomes[25,26]. The three types of interchromosomal rearrangements are reciprocal chromosome translocation (RCT), end-to-end joining (EEJ) and nested chromosome fusion (NCF)[25–27]. While RCTs do not result in descending dysploidy, both EEJs and NCFs reduce the chromosome number by one. EEJ refers not only to recombinational merger of two non-homologous chromosomes at their ends, followed by inactivation of one centromere, but also to Robertsonian translocations that create mono- or dicentric fusion chromosome along with a dispensable minichromosome or acentric fragment. Recently, this method has been applied to investigate ancient allopolyploid hybridizations in the mallow family (Malvaceae), indicating that the understanding the evolution of ancestral karyotypes may shed light on the origin of ancient hybrid genomes and their complex evolutionary relationships[26]. While the genomic analysis of the mallow family showed that mechanisms mediating chromosome number reduction align with those identified in the mustard family[5,7,10,28], the potential of the WGDI pipeline[25] for reconstructing ancestral genomes and their subsequent evolutionary trajectories has yet to be explored in the Brassicaceae.

In this work, we reconstruct the ancestral karyotypes of two lineages within the Brassicaceae family, specifically the supertribes Camelinodae (harboring *A. thaliana*, among others) and Brassicodae (harboring *Brassica* crop species). We use available chromosome-level genome assemblies from extant crucifer species and analyze their karyotype evolution by explicitly considering three types of chromosomal rearrangements (RCTs, EEJs, and NCFs), rather than merely counting the number of fission and fusion events. This approach enables us to independently establish the deep evolutionary relationships of modern crucifer genomes and to investigate the origin of two allopolyploid species with established evolutionary histories. Moreover, we examine the congruence between the shared chromosome fusion breakpoints and the boundaries of previously identified conserved genomic blocks[4,5].

## Results

### Ancestral karyotype of supertribes Camelinodae and Brassicodae (ACBK)

We followed the workflow of WGDI toolkit[25] (https://github.com/SunPengChuan/wgdi-example/blob/main/Karyotype_Evolution.md) to reconstruct ancestral karyotypes of Camelinodae and Brassicodae. All genomic data used in this study can be found in Supplementary Data 1. The *Cardamine hirsuta* genome assembly (https://www.ncbi.nlm.nih.gov/datasets/genome/GCA_963555745.1/; erroneously reported as the *C. flexuosa* genome), which proved to be superior to other assemblies (Supplementary Fig. 1), was selected for further analyses[29]. First, we identified shared chromosome-like synteny blocks (CLSBs) as the protochromosomes shared by the assembled Camelinodae genomes with different chromosome numbers ($x$ = 5, 6, 7, 8, and 16) and the Brassicodae genomes ($x$ = 7) (Supplementary Fig. 2). This resulted in the identification of the first four protochromosomes of the eight-chromosome Ancestral Karyotype of Camelinodae and Brassicodae (ACBK, Fig. 1d and Supplementary Fig. 2). We then identified two

ancestral karyotypes in Camelinodae, namely AKI and tAKI (both $p$ = 8), which share the first four chromosomes of ACBK and two additional protochromosomes (AKI5 = tAKI5 and AKI6 = tAKI6), but differ by a reciprocal translocation (abbreviated as AKI/7_8/RCT) between chromosomes 7 and 8 (Fig. 1a and d and Supplementary Fig. 3). Similarly, in the Brassicodae tribes Eutremeae, Isatideae, Schrenkielleae and Thlaspideae, we identified the ancestral karyotype, AKII ($p$ = 7), comprising the four protochromosomes of ACBK and three additional protochromosomes (Supplementary Fig. 4). Interestingly, we were able to detect protochromosomes AKI5/tAKI5, AKI6/tAKI6 and AKII5 as CLSBs in the genome assembly of *Megadenia pygmaea* ($n$ = 6, Biscutelleae) from the supertribe Heliophilodae (Fig. 1a and Supplementary Fig. 5). The identified CLSBs corroborate the phylogenetic position of the Biscutelleae[19] and Heliophilodae between the supertribes Camelinodae and Brassicodae. In relation to the AKI genome, *Megadenia* has a reciprocal translocation (AKI/7_8/RCT, blue), distinct from the AKI/7_8/RCT (orange) differentiating the AKI and tAKI ancestral genomes, as shown by their different fusion breakpoints, and a reciprocal translocation similar to AKII/6_7/RCT (Fig. 1a and Supplementary Fig. 5). In other words, the evolutionary relationships that can be inferred from current ancestral karyotypes can be represented as an unrooted tree: (((AKI, tAKI), *M. pygmaea*), AKII) (Fig. 1a). Due to reciprocal translocations defining different karyotypes, the structure of ACBK's protochromosomes remains unclear.

The directionality of chromosomal rearrangements can be determined by the chromosome structure of an outgroup taxon, much like rooted trees that use outgroups as roots. Chromosomal rearrangements induce different numbers of fusion breakpoints (BPs): EEJ involves one BP, while RCT and NCF involve two BPs. We extracted a total of 100 genes (50 on each side of a given fusion BP) as individual segments and conducted a collinearity analysis of the corresponding regions in all genomes analyzed. If the outgroup genome exhibits collinearity with a 100-gene segment spanning the corresponding fusion BP(s), these chromosomal structures are considered to be inherited from ancestral chromosomes (Fig. 1b). This allowed us to determine the sequence of reciprocal translocations and thus identify the four remaining protochromosomes of ACBK (Fig. 1c).

We used *Gynandropsis gynandra* ($n$ = 17, Cleomaceae) as an outgroup genome for comparison with the *Megadenia* genome and marked fusion BPs in the latter genome resulting from three different comparisons: AKI-tAKI, AKI-*M. pygmaea*, and AKII-*M. pygmaea* (Fig. 1c and Supplementary Figs. 3 and 5). The fusion BPs shared by *M. pygmaea* and *G. gynandra* are suggesting an ancestral karyotype similar to *M. pygmaea* and lacking three later events (AKI/7_8/RCT, blue; AKI/7_8/RCT, orange; and AKII/6_7/RCT; Fig. 1c). This was further validated by comparisons with the *Aethionema arabicum* genome (Aethionemeae, Aethionemoideae—the sister subfamily of the Brassicoideae; Supplementary Fig. 6). Next, based on the inferred directionality of ancestral chromosomal evolution, we reconstructed an ancestral karyotype of Camelinodae, namely preAKI ($p$ = 8), which included AKI5, AKI6, and two protochromosomes preAKI7 and preAKI8 (Fig. 1d). Similarly, the ancestral karyotype of Brassicodae and *M. pygmaea*, preAKII ($p$ = 7), was reconstructed based on CLSBs shared between *M. pygmaea* and AKII. Finally, a comparison of preAKI ($p$ = 8) and preAKII ($p$ = 7) revealed that they share two identical protochromosomes (preAKI6 = preAKII6, preAKI7 = preAKII5) and that the fusion of preAKI5 and preAKI8 formed chromosome preAKII7 (preAKI/5_8/EEJ) (Fig. 1d). Consequently, these protochromosomes (preAKI5–preAKI8) are the remaining four chromosomes of the eight-chromosome ACBK.

Based on the identified chromosomal rearrangements and reconstructed ancestral karyotypes, we inferred phylogenetic relationships (Fig. 2), which are in good agreement with the topology suggested by Hendriks et al.[3]. ACBK first evolved into AKI through the ACBK/7_8/RCT event. The structure of AKI ancestral genome remained conserved in the Cardamineae, whereas it was altered by a reciprocal

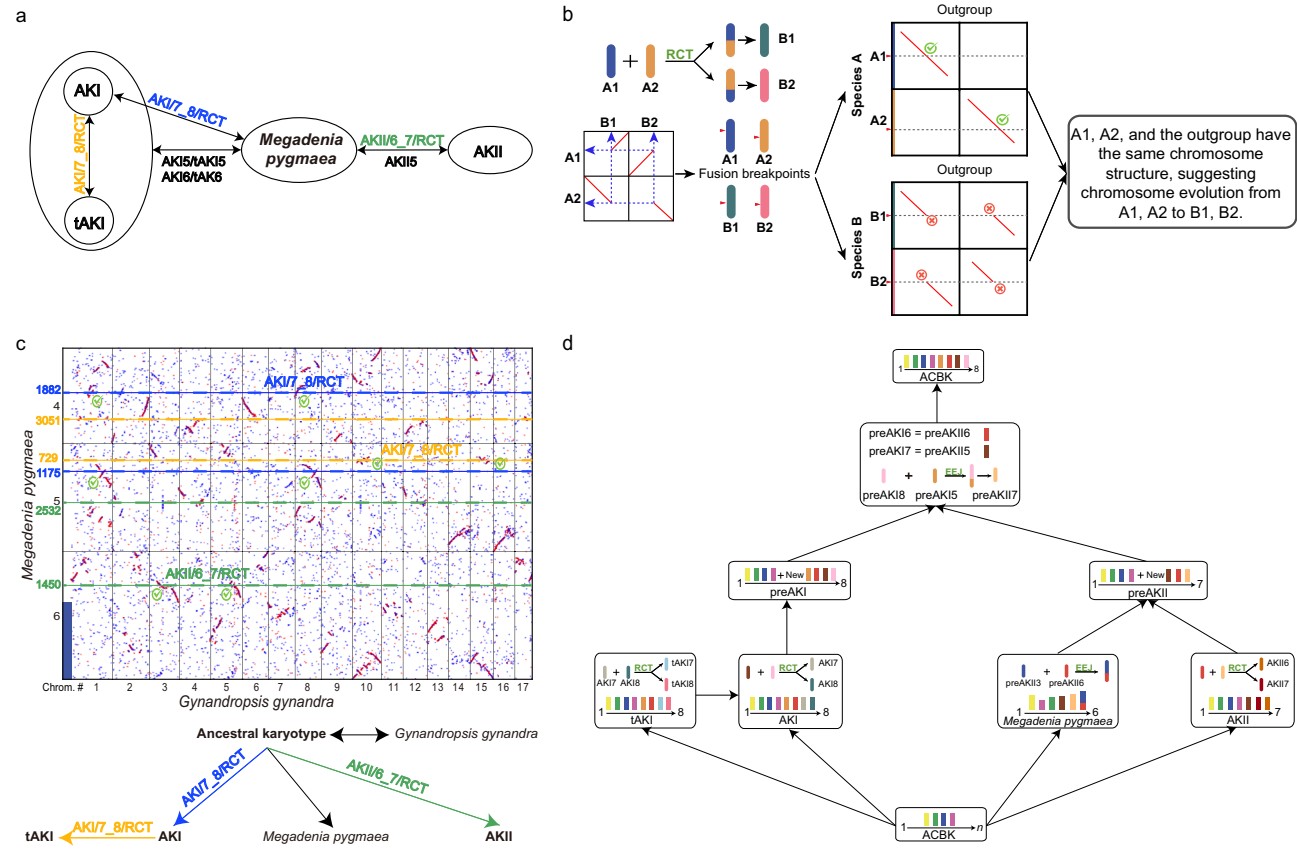

**Fig. 1 | The method and an example of incorporating an outgroup genome to reconstruct ancestral karyotypes. a** The potential transitional relationships between ancestral karyotypes within the supertribes Camelinodae (AKI and tAKI) and Brassicodae (AKII) and their relationship to *Megadenia pygmaea*. The bidirectional arrows indicate uncertainty about the direction of evolution. Chromosomal rearrangements involve the same chromosomes but are not shared by different lineages or species, as marked by different colors. Other chromosomes are shared between species and lineages as chromosome-like synteny blocks (CLSBs). **b** Identifying the fusion breakpoints of chromosomal rearrangements and comparing the corresponding chromosome structures with outgroup species allows inference of the evolutionary direction. Green ticks indicate collinearity support of the fusion events, while red crosses indicate a lack of support. **c** Comparison of fusion breakpoints between *M. pygmaea* ($n = 6$) and *Gynandropsis gynandra* ($n = 17$, Cleomaceae) elucidates the direction of chromosome fusion events. Green checkmarks indicate chromosomal structures shared between *Megadenia* and *Gynandropsis*, while arrows illustrate the evolutionary directions of the fusion events. **d** The reversal of karyotype evolution (compared to the directionality in this figure **c**) led to the identification of four additional protochromosomes of ACBK and three of preAKII, in addition to the initial four protochromosomes of ACBK. ACBK Ancestral Karyotype of Camelinodae and Brassicodae, AKII Ancestral Karyotype of Brassicodae, preAKI precursor of AKI, preAKII precursor of AKII.

translocation (AKI/7_8/RCT) towards the tAKI genome preceding the diversification of most Camelinodae tribes (e.g., Arabidopsideae, Camelineae, Cruchimalayeae, Erysimeae). Earlier studies[4,5] suggested that tAKI, known as ACK, was the most ancestral karyotype of Camelinodae (LI), with the translocation AKI/7_8/RCT (orange; equal to AK6/8 and AK8/6 chromosomes) regarded as a more recent, Cardamineae-specific event[14]. However, by adopting an outgroup approach, we challenge the original concept and have accurately reconstructed the evolutionary trajectory of ancestral karyotypes in the Camelinodae.

## Descending dysploidy in Camelinodae and Brassicodae

We have closely examined the relationship between chromosome number variation and chromosome fusion events in Camelinodae and Brassicodae species analyzed (Fig. 2). While the diploid and tetraploid Cardamineae species have retained the ancestral AKI genome (Supplementary Fig. 7), the genomes of the remaining Camelinodae tribes (Arabidopsideae, Boechereae, Camelineae, Cruchimalayeae and Erysimeae) are derived from the ancestral tAKI genome formed by a reciprocal translocation (AKI/7_8/RCT) (Supplementary Figs. 8 and 9). Both *Boechera* species analyzed have seven chromosomes due to an EEJ (tAKI/(3_8/RCT)_5/EEJ) (Supplementary Fig. 10). An independent descending dysploidy (tAKI/4_6/EEJ) from $n = 8$ to $n = 7$ occurred in the ancestry of the genus *Camelina* (Camelineae; Supplementary Fig. 11).

Subsequently, this ancestral chromosome underwent two unique reciprocal translocation events involving tAKI5, resulting in the formation of two infrageneric subclades (Supplementary Fig. 12). In *Camelina neglecta*, another EEJ (tAKI/2_8/EEJ) further reduced the number of chromosomes from 7 to 6 (Supplementary Fig. 13). The most extensive reduction in chromosome number in the diploid crucifer genomes was most likely associated with the origin of *A. thaliana* (Arabidopsideae), where three EEJ events reduced chromosome number from $n = 8$ to $n = 5$ (Supplementary Fig. 14), whereas other *Arabidopsis* species (e.g., *A. arenosa*, *A. halleri*, and *A. lyrata*) retained the ancestral chromosome number and the tAKI genome (Supplementary Fig. 8). Similarly, *Crucihimalaya lasiocarpa* (Crucihimalayeae) maintained the 8-chromosome tAKI genome altered by a single reciprocal translocation (tAKI/1_8/RCT, Supplementary Fig. 15).

Whereas some Brassicodae species retained the ancestral preAKII karyotype ($p = 7$)[7], the most recent common ancestor of the four tribes analyzed here (Eutremeae, Isatideae, Schrenkielleae, and Thlaspidae) underwent a reciprocal translocation between chromosomes 6 and 7 (preAKII/6_7/RCT), resulting in the formation of the AKII genome ($p = 7$) shared by these tribes (Supplementary Figs. 16 and 17). In the Heliophilodae, *M. pygmaea* (Biscutelleae) experienced a preAKII/3_6/EEJ event, reducing its chromosome number from 7 to 6 (Supplementary Fig. 18).

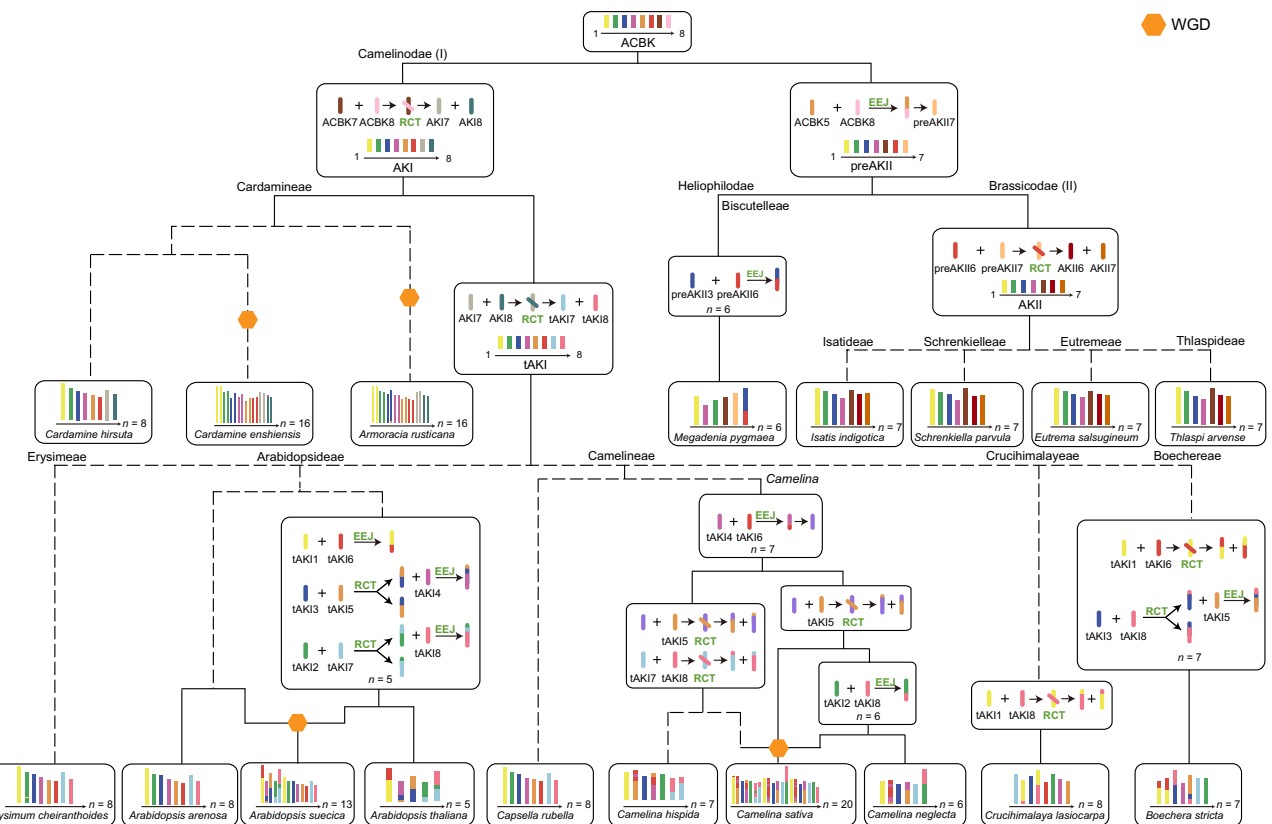

**Fig. 2 | Reconstructed karyotype evolution in three Brassicoideae supertribes (Brassicodae, Camelinodae and Heliophilodae).** Ideograms are based on the published chromosome-level genome assemblies (Supplementary Data 1) and their origins reconstructed using the WGDI[25]. Schematic phylogenetic relationships follow the phylogenetic analyses by Hendriks et al.[3]. Whole-genome duplications

are marked by orange hexagons. ACBK Ancestral karyotypes of Camelinodae and Brassicodae, AKI Ancestral Karyotype of Camelinodae (LI), tAKI translocation AKI, preAKI precursor of AKI, AKII Ancestral Karyotype of Brassicodae (LII), preAKII precursor of AKII, RCT reciprocal chromosome translocation, EEJ end-to-end joining.

## Application of shared fusion events to determine the parentage of allopolyploid species

The subgenome composition of the allotetraploid *Arabidopsis suecica* ($2n = 26$) and the allohexaploid *Camelina sativa* ($2n = 40$) has been established in several studies[30–39]. We tested how the parental genomes of allopolyploid species can be identified based on unique genome-specific chromosome fusion events. The *A. thaliana* genome ($n = 5$) underwent three EEJs and two RCTs during its origin from the tAKI genome ($p = 8$; see above and Figs. 2 and 3a; Supplementary Fig. 14). In the tetraploid *A. suecica* genome ($n = 13$), all three *A. thaliana*-specific chromosome fusions were identified in the 5-chromosome subgenome (Fig. 3a and Supplementary Fig. 14), whereas the 8-chromosome subgenome has the tAKI genome structure (Supplementary Fig. 19). Furthermore, based on the subgenome tree constructed from collinear genes (Supplementary Fig. 20), we concluded that *A. arenosa* was most likely the donor of the 8-chromosome subgenome, which is consistent with previous conclusions[30].

A similar pattern is observed in *C. sativa* ($n = 20$) (Figs. 2 and 3b). *C. sativa* shares two unique chromosomal rearrangements [tAKI/(4_6/ EEJ)_5/RCT and tAKI/7_8/RCT] with the diploid *C. hispida* ($n = 7$) and a single unique tAKI/6_8/EEJ fusion with the diploid *C. neglecta* ($n = 6$) (Fig. 3b and Supplementary Figs. 12 and 21). Moreover, the hexaploid genome contains two copies of the unique tAKI/(4_6/RCT)_5/RCT chromosome rearrangement contributed by the *C. neglecta* and *C. neglecta*-like ($n = 7$) genome (Supplementary Fig. 21). These rearrangements indicate that *C. sativa* is an allohexaploid with three distinct subgenomes derived from ancestors of *C. hispida* ($n = 7$), *C. neglecta* ($n = 6$) and *C. neglecta*-like genome ($n = 7$). This conclusion is further supported by the subgenome trees (Supplementary Fig. 20). In

contrast to previous approaches[34,37] identifying parental genomes based on the structure of all chromosomes, our method targets only the fusion BPs in the putative parental genomes and the hybrid genome.

## Breakpoints of specific chromosomal rearrangements show association with CGB boundaries and stable inheritance

To explore the correlation between the origin of the 22 CGB boundaries[5] and fusion events, we mapped both the boundaries and the fusion BPs of shared events from Fig. 2 onto the ACBK (Fig. 4a). The 22 CGBs delineate 16 boundaries (Supplementary Fig. 22), while the fusion events pinpoint 30 fusion positions (Supplementary Data 2 and 3). A comparative analysis revealed that 11 of these positions completely overlap. The incomplete coverage of fusion BPs within CGB boundaries may be due to structural variation, such as inversions, as well as non-structural factors like gene loss, which can disrupt synteny blocks. Additional sampling of chromosome-level genomes from other supertribes may identify new fusion BPs that overlap with the remaining CGB boundaries.

To assess the stable inheritance of these fusion BPs, we added 15 new genomes in our analysis, including three from Arabodae, one from Hesperodae, one from Aethionemeae, and included ten genomes previously used to reconstruct ACBK. We automated the analysis of evolutionary relationships among 33 species exclusively using fusion BPs (Fig. 4b). Notably, all Brassicaceae species, except for the tetraploid *Lepidium didymum* ($n = 16$), have predominantly inherited unique fusion events, with no instances of the repeated origin of a particular fusion BP (Figs. 2 and 4b). As the Lepidieae is an early diverging clade in Camelinodae[3], further study is needed to confirm

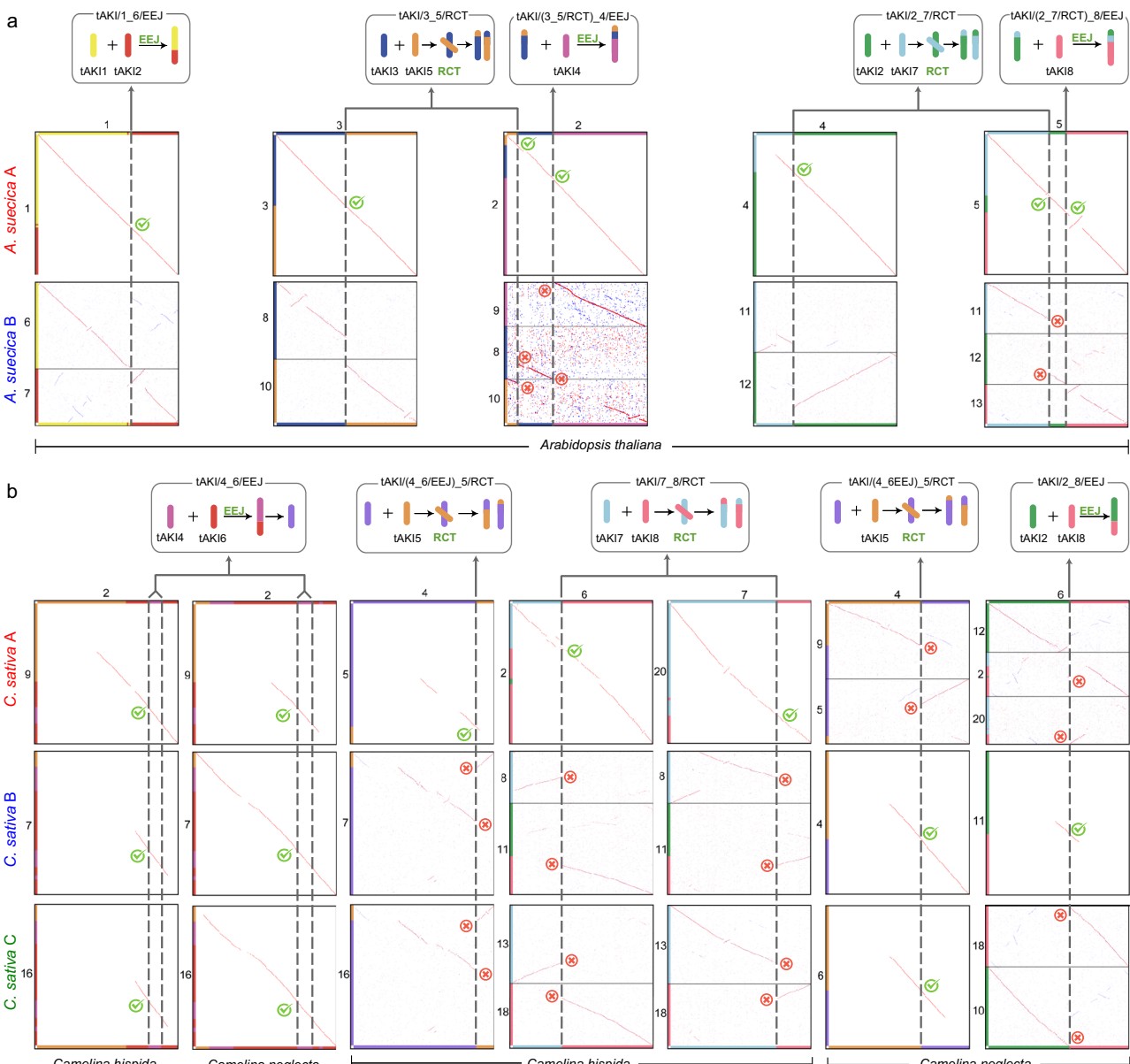

**Fig. 3 | Shared chromosome fusion events allow the subgenome identification in allopolyploid genomes. a** *Arabidopsis suecica* (*n* = 13, AB). The allotetraploid origin of *A. suecica* is confirmed by five unique chromosome fusions shared between subgenome A (*n* = 5) and *A. thaliana*, whereas no fusions are shared between *A. thaliana* and subgenome B (*n* = 8). Green ticks indicate collinearity support of the fusion events, while red crosses indicate a lack of support. **b** A chromosome fusion event (EEJ) shared between subgenomes A, B, and C of *Camelina sativa* (*n* = 20), as well as by *C. hispida* and *C. neglecta*, indicate their common ancestry. Subgenome A shares two unique events [tAKI(4_6/EEJ)_5/RCT, tAKI/7_8/RCT] with *C. hispida*, subgenome B shares two (tAKI(4_6/EEJ)_5/RCT, tAKI/2_8/EEJ) with *C. neglecta*, and *C. sativa* subgenome C shares one translocation (tAKI(4_6/EEJ)_5/RCT) with the *C. neglecta*-like genome, confirming the allohexaploid status of *C. sativa*. The two reciprocal translocations (tAKI(4_6/EEJ)_5/RCT) are distinct due to different fusion breakpoints (Supplementary Fig. 12).

the chromosomal rearrangements in *Lepidium* genomes (see the next section for details). In the meso-octoploid *Heliophila variabilis* (*n* = 11, Heliophileae, Heliophilodae), only one copy of the (ACBK/5_8/EEJ = preAKI/5_8/EEJ) event was identified, aligning with previous study suggesting that this ancient polyploid contains at least three distinct ancestral genomes[21]. The three newly added outgroups (Arabodae, Hesperodae, and Aethionemeae) do not share all of the fusion BPs. For the allopolyploid species (Fig. 3), the different copy numbers of fusion events also reflect the ancient origin of these hybrid genomes, although this is not as clearly visible as in Fig. 2. Nonetheless, this approach represents a feasible and promising strategy to infer deep evolutionary relationships based on shared chromosome fusion events, in contrast to conventional DNA-based alignment methods.

**Reconstructing evolutionary relationships of the tribe Lepidieae**

The Lepidieae, a tribe of the supertribe Camelinodae, is characterized by a high incidence of hybridization and polyploidy, and its phylogenetic position in the supertribe alternates with the Cardamineae due to nuclear-cytoplasmic conflicts[3]. Comparing *Lepidium didymum*, which underwent an independent whole-genome duplication, with ACBK, we identified two distinct chromosomal rearrangements: ACBK/5_6/RCT (one genomic copy) and ACBK/7_8/RCT (two genomic copies) (Fig. 5a). This suggests that ACBK/7_8/RCT predated the polyploidization in *L. didymum*. Importantly, the ACBK/7_8/RCT is distinct from the ACBK/7_8/RCT (blue) specific to AKI and ACBK, as evidenced by their different fusion BPs (Figs. 1 and 5b). These findings point to ancestral chromosomes ACBK7 and ACBK8 as rearrangement hotspots

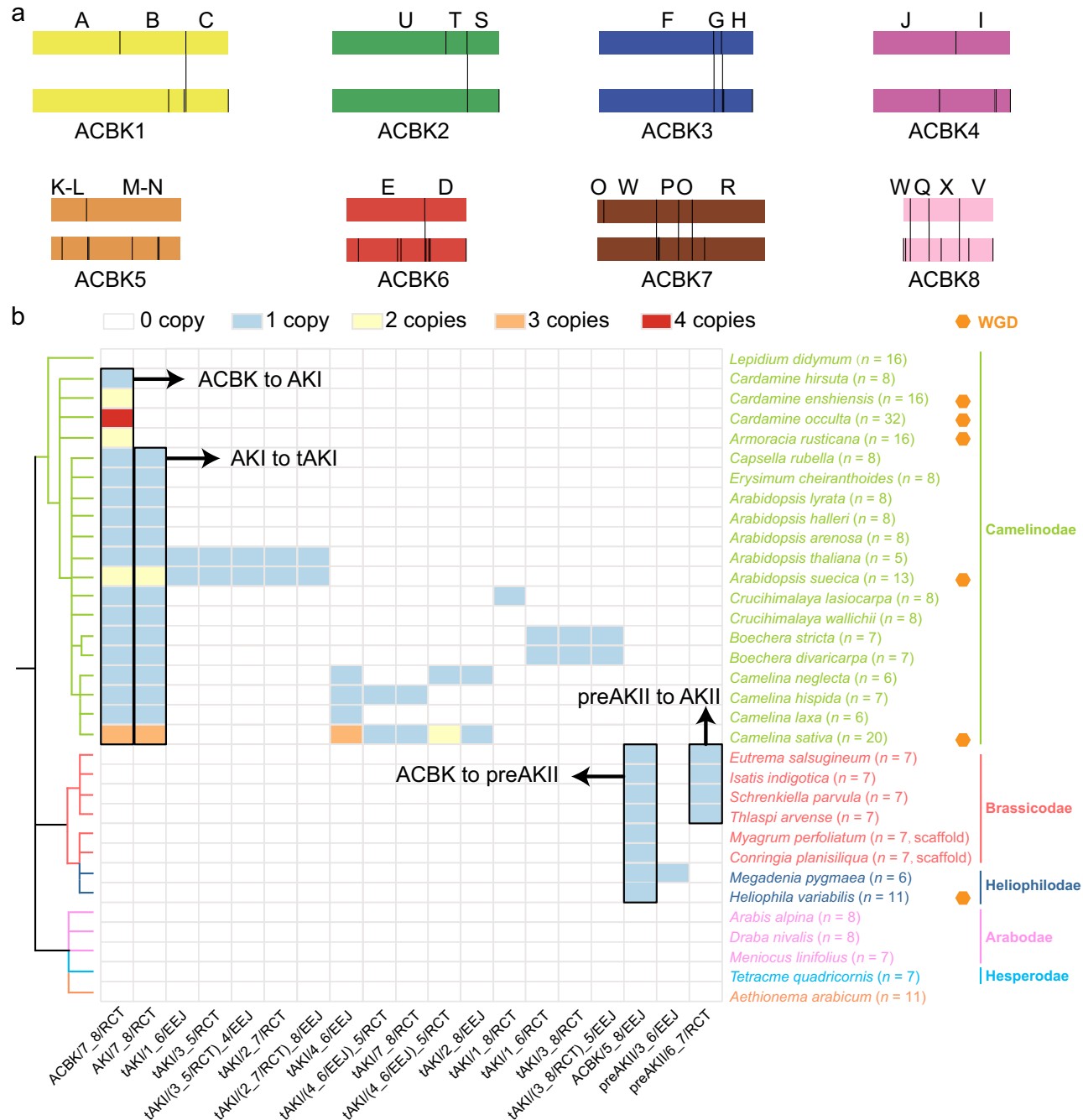

**Fig. 4 | Association of chromosomal breakpoints with conserved genomic blocks and their phylogenetic informativness. a** Comparison between conserved genomic blocks and fusion breakpoints. Rectangles of the same color denote identical ancestral chromosomes. The upper vertical lines mark the 16 boundaries of 22 conserved genomic blocks (A–X) mapped to ACBK breakpoints, while the lower lines indicate the 30 breakpoints of shared fusion events in Fig. 2, also mapped to ACBK. Vertical lines connect breakpoints separated by no more than 50 genes. **b** Supertribe-level clades inferred exclusively through fusion breakpoints. The chromosome fusion copy number variation is color-coded. Whole-genome duplications are marked as orange hexagons. *Conringia planisiliqua* and *Myagrum perfoliatum* are scaffold-level genome assemblies. ACBK Ancestral karyotypes of Camelinodae and Brassicodae, AKI Ancestral Karyotype of Camelinodae (LI), tAKI translocation AKI, preAKI precursor of AKI, AKII Ancestral Karyotype of Brassicodae (LII), preAKII precursor of AKII. Source data are provided as a Source Data file.

preceding the early cladogenesis within the Camelinodae and suggest a sister relationship between the tribes Cardamineae and Lepidieae[3].

## Discussion

We have reconstructed the ACBK as the ancestral karyotype of the analyzed genomes of supertribes Camelinodae (former Lineage I) and Brassicodae (former Lineage II) as well as of *M. pygmaea* (Biscutelleae,

supertribe Heliophilodae). Furthermore, we have clarified the evolutionary relationship between the Cardamineae and the bulk of Camelinodae tribes as well as between tribes Cardamineae and Lepidieae based on the shared ACBK genome and similar but different fusion breakpoints. Chromosome fusions with unique breakpoints may have contributed significantly to speciation and cladogenesis in the Brassicaceae, as most tribes have their unique chromosome fusion events (Fig. 2).

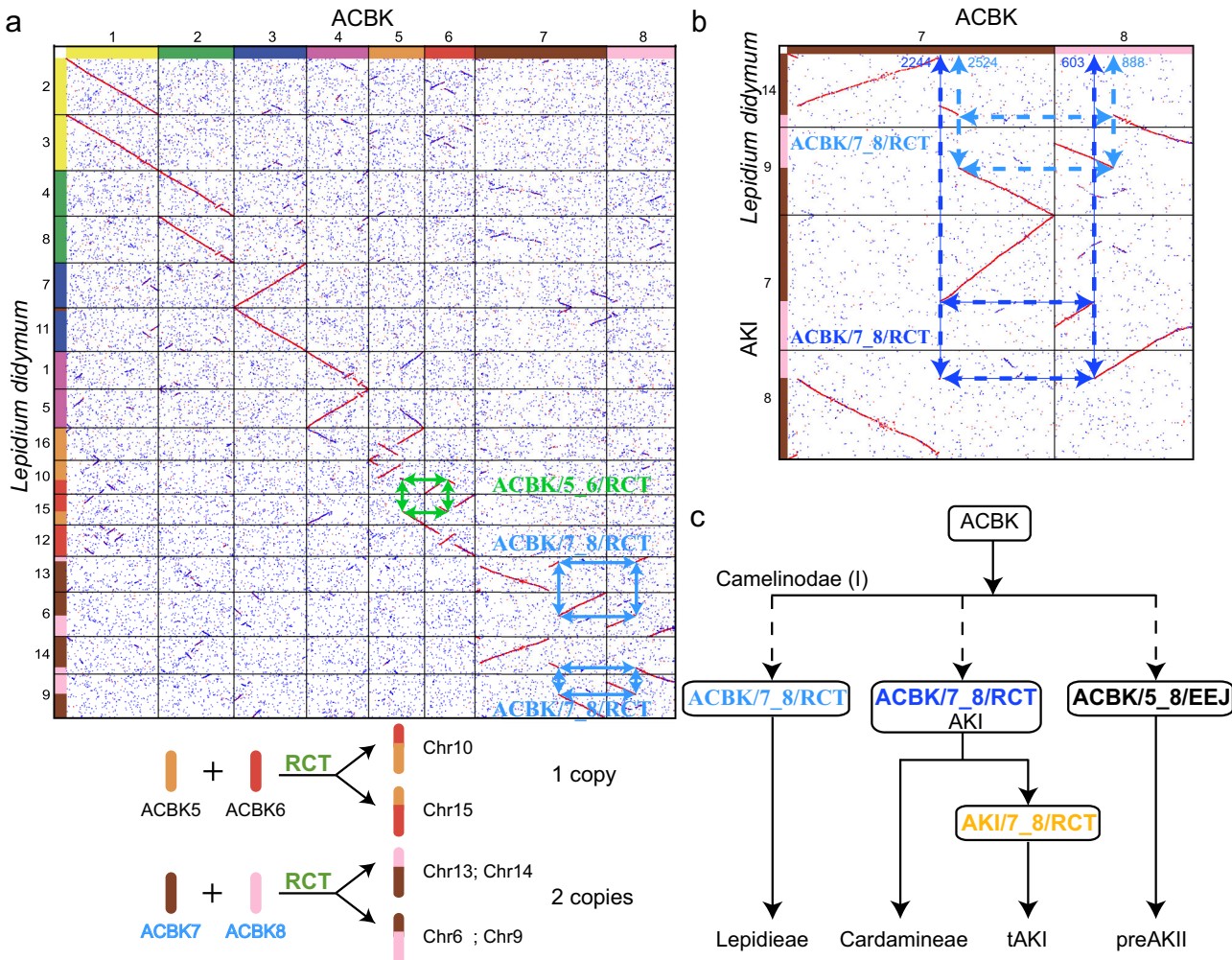

**Fig. 5 | The evolutionary relationship between the Cardamineae and Lepidieae elucidated using ancient chromosomal rearrangements. a** Chromosomal rearrangements inferred from the ancestral karyotype of supertribes Camelinodae and Brassicodae (ACBK) to *Lepidium didymum* (*n* = 16, Lepidieae). RCT reciprocal chromosome translocation. **b** The *n* = 16 genome of *L. didymum* and *p* = 8 AKI genome were shaped by similar reciprocal translocations involving the same ACBK protochromosomes, but with different breakpoints. **c** Evolutionary scheme showing the sister relationship between the Cardamineae and Lepidieae based on comparison of inferred ancestral genomes (ACBK, AKI, tAKI, and AKII) and clade-specific chromosome fusion breakpoints.

Our approach identifies CLSBs as ancestral chromosomes by pairwise comparisons between species and determines shared chromosomal rearrangements to independently establish deep evolutionary relationships between all species studied. We determine shared chromosomal rearrangements by assessing whether the genes on either side of a fusion breakpoint lie within a single synteny block, which significantly increases the uniqueness of chromosomal rearrangements. Previously identified CGBs were mainly determined based on the collinearity of homeologous chromosomes and chromosomal rearrangements[4,5,40]. We conclude that the CGB boundaries largely overlap with fusion breakpoints (Fig. 4a and Supplementary Data 3). Moreover, we can determine the evolutionary relationships between species based on the sequence and different combinations of fusion breakpoints. The clear evolutionary trajectory of fusion events, best illustrated by the sequence of rearrangements from ACBK to AKI and then to tAKI in the Camelinodae, provides a more informative representation of the evolutionary relationships between genomes and taxa. As a result, our approach offers a significant improvement over previous strategies[5,7]. In addition, other studies of karyotype evolution in plants have treated shared contiguous ancestral regions as CGBs or protochromosomes and simulated changes in chromosome number as fusion and fission events[40,41]. However, they have rarely traced shared chromosomal rearrangements or positioned fusion breakpoints in phylogenetic trees. In contrast, our method uses CLSBs as protochromosomes and tracks their dynamic evolution by accounting for shared fusion breakpoints.

Fusions and fissions are widely recognized as common events in karyotype evolution and many studies documented their high frequency[40–43]. However, our results show that all modern crucifer genomes we examined can be traced back to the ancestral karyotype through three types of interchromosomal rearrangements, without considering fusions and fissions (Fig. 2). Our results show a direct correlation between changes in chromosome number and the specific chromosome fusion mechanism: EEJ. Although nested chromosome fusions (NCFs) also reduce chromosome number by one[27], no NCF events were detected in this study. Apart from the species of the Brassicaceae and Malvaceae families[26], changes in chromosome numbers in *Vitis*[44] and Osteichthyes[45] are also consistent with the three types of interchromosomal rearrangements. Furthermore, a growing body of research suggests that fusions, not fissions, are the primary driver of chromosome number changes[24,46–48].

The construction of evolutionary relationships between studied species should be supported by shared chromosomal structural variation[26,44]. However, complex chromosomal rearrangements frequently result in fragmentation of CGBs into numerous short segments, which makes the identification of shared genomic structures considerably more difficult. In this context, collinearity detected at fusion breakpoints may provide direct evidence of chromosomal rearrangements inherited by common descent. As illustrated in Fig. 4b, this approach is likely to be crucial in future efforts to reconstruct karyotype evolution. This method is useful to address discrepancies between phylogenetic gene trees based on different genomic data. However, when the genomes to be compared have conserved karyotypes, such as those in the tribe Cardamineae (Fig. 2), it is impossible to infer phylogenetic relationships using this approach.

In summary, the application of the WGDI pipeline as demonstrated here provides a theoretical basis for family-wide ancestral genome reconstructions based on high-quality genome assemblies at the chromosome level.

## Methods

### Ancestral karyotypes, karyotype evolution and phylogenetic relationships
We applied the workflow of WGDI toolkit[25] (https://github.com/SunPengChuan/wgdi-example/blob/main/Karyotype_Evolution.md) to identify protochromosomes and to reconstruct ancestral genomes. In addition, we compared gene sequence at the fusion breakpoints with those in the outgroup genomes to identify protochromosomes and reconstruct the directionality of chromosomal rearrangements.

We used the WGDI toolkit[25] with the '-km' parameter to map the inferred eight protochromosomes of ACBK onto the chromosomes of the sample genomes. Additionally, we used the parameter '-sf' to rapidly identify chromosome fusions and record the shared fusions along with corresponding fusion breakpoints. We used WGDI with the parameter in '-fpd' to extract the dataset of fusion breakpoints and then used the parameter '-fd' to detect the presence of these fusion events in other genomes. More detailed examples can be found in Github (https://github.com/SunPengChuan/Ancestral_Brassicaceae_Karyotype/blob/main/karyotype_evolution_example/Shared_fusion_breakpoints.md). Finally, we inferred phylogenetic relationships between species and higher-order taxa by examining which genomes contain shared fusion breakpoints.

### Subgenome-aware analyses
To explore the evolutionary relationships among subgenomes in polyploid species, we phased chromosomal blocks into distinct subgenomes and constructed a subgenome phylogenetic tree. We used the WGDI[25] toolkit for both subgenome phasing and tree construction. Initially, synteny blocks were identified using the '-icl' parameter to detect collinear regions, with the most recent ancestral karyotype serving as the reference genome. Next, the '-km' parameter was applied to map protochromosomes onto the target genome. To refine subgenome assignments, we assessed the complementarity of collinear regions, structural consistency, and gene retention levels. Collinear genes were extracted using the '-a' parameter in WGDI, and phylogenetic trees for individual sets of collinear genes were generated using the '-at' parameter. These individual trees were subsequently integrated into a comprehensive subgenome phylogenetic tree using the ASTER[49] software.

To ensure the reliability of subgenome classification, we implemented an iterative optimization strategy. Whenever the tree topology conflicted with expected polyploidization events, the classification scheme was iteratively refined by re-evaluating collinear block assignments and phylogenetic relationships to ensure alignment with established evolutionary patterns. We applied the example at Github (https://github.com/SunPengChuan/wgdi-example/blob/main/phase_subgenomes.md) to phase subgenomes and to construct subgenome trees.

### CGB boundaries on the protochromosomes of ACBK
Previous studies have proposed the concept of 22 conserved genomic blocks[4,5]. We mapped 22 CGBs with different colors to the eight protochromosomes of ACBK using the WGDI toolkit[25] with the parameter '-km'. This approach allowed us to determine the position of the CGBs on the ACBK protochromosomes.

### Reporting summary
Further information on research design is available in the Nature Portfolio Reporting Summary linked to this article.

## Data availability
Data supporting the findings of this work are available within the paper and its Supplementary Information files. A reporting summary for this article is available as a Supplementary Information file. The ancestral karyotypes and phylogenetic data have been deposited in GitHub (https://github.com/SunPengChuan/Ancestral_Brassicaceae_Karyotype). In addition, examples demonstrating how to verify which fusion breakpoints are present in a given species can be found at GitHub (https://github.com/SunPengChuan/Ancestral_Brassicaceae_Karyotype/blob/main/karyotype_evolution_example/Shared_fusion_breakpoints.md). Source data are provided with this paper.

## Code availability
Custom code and pipelines for karyotype evolution are publicly available at GitHub (https://github.com/SunPengChuan/wgdi-example/blob/main/Karyotype_Evolution.md).

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

## Acknowledgements

This work was supported equally by Science and Technology Projects of Xizang Autonomous Region, China (XZ202402ZD0005) and Natural Science Foundation of China (32030006). The work was also supported by the Czech Science Foundation (21-03909S) and the project TowArds Next GENeration Crops (CZ.02.01.01/00/22_008/0004581) of the ERDF Programme Johannes Amos Comenius.

## Author contributions

P.S. led and coordinated the project. P.S., M.A.L., and J.L. designed analyses. X.J., D.M., L.X., X.L., Q.H., and X.S. analyzed Brassicaceae genomes. P.S. and X.J. finished karyotype and phylogenetic analyses. P.S., M.A.L., I.A.A., and J.L. wrote and edited the manuscript. All authors read and approved the final version of the manuscript.

## Competing interests

The authors declare no competing interests.

## Additional information

Jianquan Liu, Martin A. Lysak or Pengchuan Sun.

