## [Peer Review file · Nature Communications]

Chromosome fusions shaped karyotype evolution and evolutionary relationships in the model family Brassicaceae

Corresponding Author: Professor Martin Lysak

Version 0:

Reviewer comments:

Reviewer #1

(Remarks to the Author)

In this manuscript, the authors used chromosome-level genome assemblies to add to previous studies that have defined the Ancestral Crucifer Karyotype (ACK) in the plant family Brassicaceae. Previous studies have identified 22 conserved genomic blocks (CGBs), and in this study they look at chromosome fusion breakpoints of reciprocal translocations and end-to-end joinings from representative species.

They found eight protochromosomes representing the common ancestral karyotype (ACBK) of the two Brassicaceae supertribes: Camelinoideae (Lineage I, which includes Arabidopsis) and Brassicoideae (Lineage II, which includes Brassica crops).

I have no problems with the methods and main results as described (no flaws or overstatements):

Descending dysploidy in Camelinoideae and Brassicoideae (Fig 2);

Application of shared fusion events to determine the parentage of allopolyploid species (Fig 3); and

CGB boundaries appear to correlate with the breakpoints of specific chromosomal rearrangements, and these breakpoints exhibit stable inheritance (Fig 4).

There appears to be enough detail provided in the methods for the work to be reproduced.

My main criticism of the manuscript is that the results are not particularly noteworthy or transformative. The main discussion points have been made in the literature previously, namely that

(1) Chromosomal rearrangements/structural variation may have contributed to speciation/cladogenesis; (2) chromosomal block boundaries overlap with fusion breakpoints;

(3) subset of three types of chromosomal rearrangements can be traced back on a phylogeny of the mustard family (some new details added on reciprocal chromosome translocation, end-to-end joining, and nested chromosome fusion); and

(4) the WGD pipeline is just one approach to ancestral genome reconstruction.

In summary, the authors used previously published genomes and applied a new comparative genomics workflow to further fill in details of karyotype evolution in the Brassicaceae; however, the advance is more incremental than novel (at least as currently written).

Reviewer #2

(Remarks to the Author)

The manuscript is of high quality and was a pleasure to read. It covers a very interesting and timely topic, by providing exciting new experimental data to interpret karyotype evolution in Brassicaceae. I have only some minor comment, directly annotated in the manuscript.

In particular, I recommend to pay attention to the use of n , x , or p where appropriate.

Reviewer #3

(Remarks to the Author)

The manuscript entitled "Chromosome fusions shaped karyotype evolution and evolutionary relationships in the model family Brassicaceae" reconstructed the ancestral karyotypes of Camelinoideae and Brassicoideae supertribes, investigated their subsequent evolution including chromosome fusions, and confirmed that these fusion events elucidate the evolution of chromosome numbers and serve as phylogenetically informative genomic signatures. The article is focused on the very important aspect of chromosome evolution in Brassicaceae ($x=6, 7, 8$) by means of sorting the fragment rearrangements (they are concisely present in Figures), while the current edition format of main text is difficult for reader understanding, decision needs to be drawn after major revision and add the convincing experimental evidence. For the manuscript improvement, below points are suggested for authors.

1 In the first paragraph of result section, there are 37 abbreviations, it is many abbreviations to interrupt the readers' understanding. The similar problem readers met in the paragraph of lines 158–177.

2 In Fig. 1, the meanings of the circled check and cross signs need denoted in the legend. The same legend is need for Fig. 3.

3 In Fig. 4a, the legend showed the shared fusion events in Fig. 2. Could authors elucidated the meaning of eight pairs of the colourful squares in Fig. 4a in order to realize the self-explanatory. What are the meanings for the abbreviations (e.g., ACBK, AKI, tAKI in the legends although they had ever been elucidated in main text somewhere) either in Fig. 4a and b or in the legend. Among the black vertical bars, which parts represented the 30 breakpoints of shared fusion event in Fig. 2, request authors labeled out in order to distinguish with those labeled by letters "A to W". There are 23 capital letters between A to W, what exact means for the sentence, i.e., 16 boundaries of 22 conserved genomic blocks, could authors relabel or reorganized legend in order to show Fig. 4 results understandable, which is critical important to grasp the important findings from the synteny analysis result of the manuscript. One question for authors, how to reduce the influence of genome assembly quality for the chromosome fusion copy number prediction (0 to 4 copies of the fusion copies) in Fig. 4b cause only two scaffold-level genomes together with 31 remaining genomes of unscaffold-level.

4 In Table S2, the fusion breakpoints for chromosomal rearrangement, fusion breakpoints list, the number 1984, 653, 2548, I can't find the corresponding base start and end location or the corresponding explanation database (either on the website https://github.com/SunPengChuan/wgdi-example/blob/main/Karyotype_Evolution.md, this website data is easy to followed by the way appreciation for generators' endeavour). If it is the authors research group naming system (the corresponding database how to generate a clarified version at least to trace the breakpoints listed in table S2 and S3), which is need public available during review. Chromosome fusion breakpoints in Table S3 are also need public available.

5 In lines 271-272, "further study is needed to confirm the chromosomal rearrangements in *Lepidium* genomes", according to the FISH technology to tracing the single-copy fragment localization, the unique fusion events can be visualized.

6 For chromosome-specific fusions were very interested by other researchers, could authors summarize the investigated species in supplementary table format to show related start and end base information for chromosome-specific fusions present in Fig. 2 (labeled with signs to show which is chromosome-specific fusions). As the elucidation in lines 307-309, they may be contributed significantly to speciation or lineage divergence for other researchers go further research, how to make a solution to so many abbreviations they are disturbed the smoothly understand for such important results, request authors make a clear presentation of the main text later.

7 In line 324, what is the previous strategies, could authors list the advantages of the current strategies compared with the previous one.

8 In line 335-336, fusions not fissions are the primary driver of chromosome number changes, if FISH experimental evidence absence from current research scope, at least the base fusion point evidence to support the discussion points in order to convince the fusion point is the real one not the poor assembly caused. Especially, the below paragraph, the fusion breakpoints provide direct evidence for chromosome rearrangements. So highly convincing fusion point visualization base-by-base evidence is necessary together with synteny evidence in the manuscript.

Version 1:

Reviewer comments:

Reviewer #1

(Remarks to the Author)

I have no additional comments on the revised manuscript

Reviewer #2

(Remarks to the Author)

I am happy with the revisions made by the authors after my comments/suggestions, all properly addressed. The same applies concerning the answers and modifications made after the comments of the reviewer 3.

Answers to Reviewers' comments

Authors: We would like to thank the reviewers for their constructive feedback. All three reviewers provided valuable comments and suggestions that helped us to improve our paper, and we have made an effort to address all of them.

Reviewer #1 (Remarks to the Author):

In this manuscript, the authors used chromosome-level genome assemblies to add to previous studies that have defined the Ancestral Crucifer Karyotype (ACK) in the plant family Brassicaceae. Previous studies have identified 22 conserved genomic blocks (CGBs), and in this study they look at chromosome fusion breakpoints of reciprocal translocations and end-to-end joinings from representative species. They found eight protochromosomes representing the common ancestral karyotype (ACBK) of the two Brassicaceae supertribes: Camelinoideae (Lineage I, which includes *Arabidopsis*) and Brassicoideae (Lineage II, which includes *Brassica* crops). I have no problems with the methods and main results as described (no flaws or overstatements): Descending dysploidy in Camelinoideae and Brassicoideae (Fig 2); Application of shared fusion events to determine the parentage of allopolyploid species (Fig 3); and CGB boundaries appear to correlate with the breakpoints of specific chromosomal rearrangements, and these breakpoints exhibit stable inheritance (Fig 4). There appears to be enough detail provided in the methods for the work to be reproduced.

My main criticism of the manuscript is that the results are not particularly noteworthy or transformative. The main discussion points have been made in the literature previously, namely that (1) Chromosomal rearrangements/structural variation may have contributed to speciation/cladogenesis; (2) chromosomal block boundaries overlap with fusion breakpoints; (3) subset of three types of chromosomal rearrangements can be traced back on a phylogeny of the mustard family (some new details added on reciprocal chromosome translocation, end-to-end joining, and nested chromosome fusion); and (4) the WGDI pipeline is just one approach to ancestral genome reconstruction.

In summary, the authors used previously published genomes and applied a new comparative genomics workflow to further fill in details of karyotype evolution in the Brassicaceae; however, the advance is more incremental than novel (at least as currently written).

Response 1: We appreciate Reviewer's recognition of the methods and results we have presented in our work. In addition to refining the ancestral karyotype and resolving the phylogenomic controversy surrounding the tribes Cardamineae and Lepidieae, the most novel aspect of our manuscript lies in our innovative approach to analyze and reconstruct karyotype evolution.

Most previous studies typically simulated karyotype evolution by treating contiguous ancestral regions (CARs) as protochromosomes or conserved genomic blocks, simulating or discussing changes via fusion and fission events based on phylogenetic relationships already constructed. As mentioned in our Discussion, previous studies have often reported extensive fusions and fissions; yet, such events should be easy to validate. Regardless of whether it is a fusion or a fission, the genes flanking the breakpoints - including their collinearity and homology - should show a

consistent non-random pattern (as opposed to a random gene order). **Strikingly, previous publications quantifying these events frequently neglected the shared fusion breakpoints.** Had such validations been included (as demonstrated in our Figures 3 and 4), confirming these fusions or fissions within a genus or closely related taxa would be straightforward. This observation underscores our main finding: pivotal role of chromosome fusions in reconstructing evolutionary relationships.

The conventional approach of segmenting ancestral chromosomes into conserved genomic blocks is increasingly inadequate to advance the study of karyotype evolution, because the mirrored ancestral chromosomes in present-day genomes are not entire or continuous, but fragmented with many blank regions (i.e., regions that do not show clear homeology with the ancestral genome(s)). As the number of sequenced species increases, these genomic blocks need to be further subdivided into smaller units (Liu et al., 2024, doi:[10.1016/j.xplc.2024.100878](https://doi.org/10.1016/j.xplc.2024.100878)). However, the representation of chromosomal rearrangements as colored block combinations fails to capture identical fusion events or reflect the directional aspects of the rearrangements. In contrast, our method, as shown in Figure 2, accurately pinpoints the ancestral node at which specific events occurred, achieving a precision unattainable with block-based models.

Departing from the previous methods for ancestral karyotype construction, our karyotype reconstruction method designates chromosome-like syntenic blocks (CLSBs) as protochromosomes. By integrating three types of chromosomal rearrangements, we can trace the dynamic evolution of ancestral karyotypes and independently constructed evolutionary relationships. For example, in *Arabidopsis thaliana* and *Crucihimalaya lasiocarpa*, we illustrate how all extant genomes can be traced back to ancestral karyotypes through these three rearrangement types. Our method constructs phylogenetic relationships based on shared chromosomal rearrangements from ancestral karyotype evolution independently of other evidence, and thus, differs fundamentally from other approaches that focus solely on counting fusion and fission events. **Thus, our manuscript provides a fundamental example in the field of the reconstruction of plant karyotype evolution. With broader taxonomic validation, our approach could redefine analytical standards in this field.**

Reviewer #2 (Remarks to the Author):

The manuscript is of high quality and was a pleasure to read. It covers a very interesting and timely topic, by providing exciting new experimental data to interpret karyotype evolution in Brassicaceae. I have only some minor comment, directly annotated in the manuscript.

In particular, I recommend to pay attention to the use of n, x, or p where appropriate. **Response 1:** Following this suggestion, we have corrected the usage of 'n', 'x', or 'p' where appropriate.

Line 57-58: Very interestingly n = 8 is also the second-best inferred ancestral chromosome number in Brassicaceae reported in Table S1 Carta et al. (2020, already cited by you), using a totally different approach just considering the chromosome number variation with Chromevol: 7=0.590, 8=0.340, 6=0.035

Response 2: Thank you for this remark. Many Brassicaceae species have chromosome numbers of 8 or multiples thereof. However, ancestral karyotype studies should prioritize tracing the dynamic evolutionary trajectories of ancestral chromosomes rather than merely determining their original number. For instance, although ACBK, AKI, and tAKI all share $p = 8$, reciprocal translocations result in distinct genomic architectures, highlighting the deeper complexities of karyotype evolution often obscured by a sole focus on numbers.

Line 66-67: here, I would suggest to use p_1 and p_2 , respectively, since as far I can argue both these numbers are inferred ancestral basic chromosome numbers. See Peruzzi (2013):

<https://www.tandfonline.com/doi/full/10.1080/11263504.2013.861533>

Response 4: We have replaced " n " with " p " without distinguishing between p_1 and p_2 (good point!).

Line 121: are these actually n (the number of chromosomes in the gametes, so that all the studied species are diploid) or x (basic chromosome number, so that both diploid and polyploid species are considered)?

Response 5: Our approach is applicable to polyploids, ' x ' is more appropriate.

Reviewer #3 (Remarks to the Author):

The manuscript entitled "Chromosome fusions shaped karyotype evolution and evolutionary relationships in the model family Brassicaceae" reconstructed the ancestral karyotypes of Camelinoideae and Brassicoideae supertribes, investigated their subsequent evolution including chromosome fusions, and confirmed that these fusion events elucidate the evolution of chromosome numbers and serve as phylogenetically informative genomic signatures. The article is focused on the very important aspect of chromosome evolution in Brassicaceae ($x=6, 7, 8$) by means of sorting the fragment rearrangements (they are concisely present in Figures), while the current edition format of main text is difficult for reader understanding, decision needs to be drawn after major revision and add the convincing experimental evidence. For the manuscript improvement, below points are suggested for authors.

1 In the first paragraph of result section, there are 37 abbreviation words, it is many abbreviations to interrupt the readers' understanding. The similar problem readers met in the paragraph of lines 158–177.

Response 6: We have reduced unnecessary abbreviations to improve readability.

2 In Fig. 1, the meanings of the circled check and cross signs need denoted in the legend. The same legend is need for Fig. 3.

Response 7: Thank you. Corrected.

3 In Fig. 4a, the legend showed the shared fusion events in Fig. 2. Could authors elucidated the meaning of eight pairs of the colourful squares in Fig. 4a in order to realize the self-explanatory.

Response 8: We have refined the description of Fig. 4a to ensure it is clear and easy to understand.

What are the meanings for the abbreviations (e.g., ACBK, AKI, tAKI in the legends although they had ever been elucidated in main text somewhere) either in Fig. 4a and b or in the legend.

Response 9: Thank you. We have added those descriptions.

Among the black vertical bars, which parts represented the 30 breakpoints of shared fusion event in Fig. 2, request authors labeled out in order to distinguish with those labeled by letters "A to W". There are 23 capital letters between A to W, what exact means for the sentence, i.e., 16 boundaries of 22 conserved genomic blocks, could authors relabel or reorganized legend in order to show Fig. 4 results understandable, which is critical important to grasp the important findings from the synteny analysis result of the manuscript.

Response 10: We acknowledge an error in our previous wording: "A to W" was incorrect. We have now revised it to "A to X," encompassing 24 capital letters. Within this range, K and L form one block (K-L), and M and N form another block (M-N), resulting in a total of 22 conserved genomic blocks. The 16 boundaries of these 22 genomic blocks are clearly mapped to their positions in ACBK, as shown in Supplementary Figure 22.

One question for authors, how to reduce the influence of genome assembly quality for the chromosome fusion copy number prediction (0 to 4 copies of the fusion copies) in Fig. 4b cause only two scaffold-level genomes together with 31 remaining genomes of unscaffold-level.

Response 11: The primary differences in chromosome fusion copy numbers stem from the ploidy levels of modern genomes compared to their ancestral karyotypes. For instance, the allotetraploid horseradish (*Armoracia rusticana*) has a copy number of 2 for ACBK/7_8/RCT (see Figure 4, first column). Similarly, *Camelina sativa*, an allohexaploid, has a copy number of 3 for ACBK/7_8/RCT. Even in scaffold-level genome assemblies, fusion breakpoints can still be accurately detected, provided they do not fall to scaffold boundaries.

4 In Table S2, the fusion breakpoints for chromosomal rearrangement, fusion breakpoints list, the number 1984, 653, 2548, I can't find the corresponding base start and end location or the corresponding explanation database (either on the website https://github.com/SunPengChuan/wgdi-example/blob/main/Karyotype_Evolution.md, this website data is easy to followed by the way appreciation for generators' endeavour). If it is the authors research group naming system (the corresponding database how to generate a clarified version at least to trace the breakpoints listed in table S2 and S3), which is need public available during review. Chromosome fusion breakpoints in Table S3 are also need public available.

Response 12: The numbers 1984, 653, and 2548 indicate the ordinal positions of this gene on the chromosomes of related species. Annotation information may vary slightly across different versions. We have recently uploaded the annotation results we used, which are available at https://github.com/SunPengChuan/Ancestral_Brassicaceae_Karyotype/tree/main/genomes. By referring to the first column (chromosome) and the sixth column (gene order) in the GFF file, the specific gene location can be determined. Additionally, we have generated a fusion breakpoint database, which is stored at https://github.com/SunPengChuan/Ancestral_Brassicaceae_Karyotype/tree/main/fusio

n_breakpoints_database. For examples how to use this database, please see https://github.com/SunPengChuan/wgdi-example/blob/main/Shared_fusion_positions.md.

The dataset extracted based on the information from Table S2 is available at https://github.com/SunPengChuan/Ancestral_Brassicaceae_Karyotype/tree/main/fusion_breakpoints_database. The information in Table S3 is similar to that in Table S2 and can be directly obtained from the GFF file of ACBK, located at https://github.com/SunPengChuan/Ancestral_Brassicaceae_Karyotype/blob/main/ancestors/ACBK/ACBK.gff.

5 In lines 271-272, “further study is needed to confirm the chromosomal rearrangements in *Lepidium* genomes”, according to the FISH technology to tracing the single-copy fragment localization, the unique fusion events can be visualized.

Response 13: In the last section of RESULTS we have elaborated on why *Lepidieae* lack the shared fusion event ACBK/7_8/RCT observed in other *Camelinodae* species (see Figure 4, first column). Although FISH analyses remain valuable, assembled chromosomes or scaffolds provide greater accuracy of the structural information and enable cross-species validation within the same genus (See **Response 16**).

6 For chromosome-specific fusions were very interested by other researchers, could authors summarize the investigated species in supplementary table format to show related start and end base information for chromosome-specific fusions present in Fig. 2 (labeled with signs to show which is chromosome-specific fusions). As the elucidation in lines 307-309, they may be contributed significantly to speciation or lineage divergence for other researchers go further research, how to make a solution to so many abbreviations they are disturbed the smoothly understand for such important results, request authors make a clear presentation of the main text later.

Response 14: Table S4 contains fusion breakpoint details linked to specific genes. We have uploaded corresponding GFF files to accurately locate these genes (see **Response 12**).

7 In line 324, what is the previous strategies, could authors list the advantages of the current strategies compared with the previous one.

Response 15: The advantages of our approach over current strategies for studying karyotype evolution are detailed in **Response 1**. We have now incorporated this description into the revised manuscript.

8 In line 335-336, fusions not fissions are the primary driver of chromosome number changes, if FISH experimental evidence absence from current research scope, at least the base fusion point evidence to support the discussion points in order to convince the fusion point is the real one not the poor assembly caused. Especially, the below paragraph, the fusion breakpoints provide direct evidence for chromosome rearrangements. So highly convincing fusion point visualization base-by-base evidence is necessary together with synteny evidence in the manuscript.

Response 16: In the WGDI pipeline, determining shared fusion breakpoints can be visualized through dot plots. For instance, in the demonstration example (https://github.com/SunPengChuan/wgdi-example/blob/main/Shared_fusion_positions.md), fusion breakpoints are marked with red and blue colors on either side (Figure S1). A shared fusion event is inferred if a

synteny block spans the fusion breakpoint (a green vertical line), as exemplified by AK1_1 and AK1_2 (Table S2). Conversely, if the syntenic regions on either side of the breakpoint correspond to distinct chromosomes or distant genomic regions (e.g., AK_17 and AK_18), the fusion is considered to be absent. Due to visual clutter from displaying all fusion breakpoints in a single figure, only representative examples are shown here. Users can obtain a comprehensive list of shared fusion breakpoints directly using the ‘-fd’ command in WGDI. Since the fusion breakpoints are determined on the basis of gene synteny, we can only provide positions at the gene level and not specific nucleotide positions (Table S2). However, this is sufficient to determine whether chromosomal rearrangements are shared.

Figure S1 | Comparison of *Capsella rubella* ($n = 8$) with the fusion breakpoint database identifies shared breakpoints.

In addition, genomes from other species within the same genus can be used to validate whether fusion breakpoints are shared by species in this genus. For instance, the genus *Lepidium* includes a scaffold-level genome of *Lepidium campestre* (accessible at: https://www.ncbi.nlm.nih.gov/datasets/genome/GCA_009757365.2/). As an example, consider the two ACBK/7_8/RCT fusion events, which correspond to the breakpoints AK1_1 and AK1_2 (from ACBK to AKI), as well as AK21_1 and

AK21_2 (from ACBK to *Lepidium*), each exhibiting distinct fusion breakpoints (see Table S2 and Figure S2). Neither *L. didymum* ($n = 16$) nor *L. campestre* ($n = 8$) possesses the fusion breakpoints AK1_1 or AK1_2. In contrast, *L. campestre* does contain the fusion breakpoint AK21_1, indicating that this breakpoint was likely present in the common ancestor of these two species.

Figure S2 | Comparison of two *Lepidium* genomes and identification of shared breakpoints based on the fusion breakpoint database.

For user's convenience, step-by-step operational guidelines and additional examples are now available at:
https://github.com/SunPengChuan/Ancestral_Brassicaceae_Karyotype/karyotype_evolution_example.